# Multi PILOT: Learned Feasible Multiple Acquisition Trajectories for Dynamic MRI

**Tamir Shor**                                                          TAMIR.SHOR@CAMPUS.TECHNION.AC.IL
**Tomer Weiss**                                                        TOMER-WEISS@CS.TECHNION.AC.IL
**Dor Noti**                                                                DORNOTI@GMAIL.COM
**Alex Bronstein**                                                    BRON@CS.TECHNION.AC.IL
*Computer Science, Technion, Haifa, Israel*

**Editors:** Accepted for publication at MIDL 2023

## Abstract

Dynamic Magnetic Resonance Imaging (MRI) is known to be a powerful and reliable technique for the dynamic imaging of internal organs and tissues, making it a leading diagnostic tool. A major difficulty in using MRI in this setting is the relatively long acquisition time (and, hence, increased cost) required for imaging in high spatio-temporal resolution, leading to the appearance of related motion artifacts and decrease in resolution. Compressed Sensing (CS) techniques have become a common tool to reduce MRI acquisition time by subsampling images in the $k$-space according to some acquisition trajectory. Several studies have particularly focused on applying deep learning techniques to learn these acquisition trajectories in order to attain better image reconstruction, rather than using some predefined set of trajectories. To the best of our knowledge, learning acquisition trajectories has been only explored in the context of static MRI. In this study, we consider acquisition trajectory learning in the dynamic imaging setting. We design an end-to-end pipeline for the joint optimization of multiple per-frame acquisition trajectories along with a reconstruction neural network, and demonstrate improved image reconstruction quality in shorter acquisition times. The code for reproducing all experiments is accessible at https://github.com/tamirshor7/MultiPILOT.

**Keywords:** Magnetic Resonance Imaging (MRI), fast image acquisition, image reconstruction, dynamic MRI, deep learning.

## 1. Introduction

Magnetic Resonance Imaging (MRI) has become one of the most popular medical imaging techniques. It is often favored over other technologies due to its non-invasiveness, lack of harmful radiation, and excellent soft-tissue contrast. In particular, for some tasks, *dynamic* MRI was shown to be substantially better applicable than static MRI. Such tasks include but are not limited to cardiac MRI, tissue motion, and cerebrospinal fluid (CSF) flow analysis.

A major drawback of MRI, however, is that it requires relatively long scan times. This not only makes MRI scans expensive but also requires patients to remain still for long periods of time. Aside from causing discomfort, prolonged scanning is more susceptible to the appearance of imaging artifacts originating from the patient's movement. In the setting of dynamic MRI, reducing frame acquisition time directly increases the temporal resolution and reduces the in-frame motion artifact of the organ of interest (e.g., the heart).

A popular approach for reducing scan time is Compressed Sensing (CS) techniques - these methods subsample the Fourier space ($k$-space) of the image according to some

predefined trajectory. CS is usually used in a pipeline prior to applying some reconstruction logic for the recovery of information lost in subsampling and for filtering blurring and aliasing artifacts caused by violating the Nyquist sampling criterion (Zaitsev et al., 2015).

**Previous work**   Following recent years' developments in deep learning and its applicability in inverse problem solving (Ongie et al., 2020), many recent studies opted for fixing some predefined handcrafted acquisition trajectory (e.g., Cartesian, Radial, Golden Angle, – henceforth collectively referred to as *fixed trajectories* in this paper) and focusing on developing deep learning models for denoising and restoring the image data lost in undersampling (Hammernik et al., 2018; Hyun et al., 2018), or performing super-resolution reconstruction (Chen et al., 2020b; Masutani et al., 2020).

Extensive research had also been made to design good handcrafted acquisition trajectories, both in the context of static ((Larson et al., 2007; Yiasemis et al., 2023)) and dynamic ((Utzschneider et al., 2021; Bliesener et al., 2020) MRI. Despite its crucial impact on the resulting image, *learning* the acquisition trajectories within the $k$-space has been so far studied to a much lesser extent. While trajectory optimization can be performed over a set of Cartesian subsampling schemes (Weiss et al., 2020; Bahadir et al., 2020), recent research unveiled the potential in optimizing over more general, non-Cartesian acquisition trajectories (Alush-Aben et al., 2020; Weiss et al., 2021; Wang et al., 2021; Chaithya et al., 2022). The latter case is considered more complex as the optimization procedure must impose hardware-dictated kinematic feasibility constraints that every sampling trajectory must satisfy. Without constraining the optimization, trajectories could violate these requirements and be unrealizable in real MRI machines.

Our work focuses on expanding PILOT – an end-to-end framework for joint optimization of physically feasible $k$-space acquisition trajectories and image reconstruction neural network previously introduced by (Weiss et al., 2021) in the static MR imaging setting. We extend this framework to the *dynamic* MRI setting. While one can naively adapt PILOT for dynamic MRI image reconstruction by using a single learned trajectory across multiple consecutive frames, learning distinct trajectories across the frames hides the potential for improved image reconstruction which is exploited and demonstrated in the present study.

From this perspective, dynamic MRI differs from its static counterpart in two important aspects. Firstly, in dynamic MRI, each data sample consists of some integer number $n$ of frames. This implies generalizing the trajectory learning problem to learning $n$ independent trajectories. As we later show, jointly learning $n$ feasible trajectories along with a reconstruction network is a harder optimization problem that requires non-trivial extensions of the initial pipeline and training regime presented in PILOT. Secondly, dynamic MRI data samples the images of the same organ across time, resulting in high cross-frame redundancy, which we exploit for more efficient sampling.

**Contributions**   This paper makes the following contributions:

1. We present Multi-PILOT – an end-to-end pipeline for the joint optimization of multiple per-frame feasible acquisition trajectories along with a reconstruction model capable of taking cross-frame data redundancy into consideration. We demonstrate Multi-PILOT's ability to achieve superior cross-frame image reconstruction results compared to that of PILOT (that learns a single learned trajectory for all frames)

and to that of constant trajectory-based reconstruction. Our improvement is shown to be expressed both in sampling time and reconstruction quality.

2. We present two trajectory learning-related training techniques that we refer to as *trajectory freezing* and *reconstruction resets*. While we demonstrate the contribution of these methods to reconstruction results for static and dynamic MRI using only our pipeline, these techniques are generalizable to other joint sampling-reconstruction optimization tasks.

3. Our work demonstrates the intricacy of jointly learning the acquisition trajectories and the reconstruction network. We present quantitative evidence for the shortcomings of 'naïve' learning of independent per-frame trajectories without incorporating any of the additional considerations proposed in this paper.

## 2. Methods

We adopt the approach employed by (Weiss et al., 2021) in PILOT – a pipeline consisting of a subsampling layer simulating the data acquisition, a regridding layer creating the subsampled image on a Cartesian grid, and a reconstruction layer for recovering the subsampled image. The subsampling and regridding layers are parametrized by the $k$-space acquisition trajectory coordinates which are jointly learned with the reconstruction parameters in order to find the optimal trajectory. In the remainder of this section, we present how each layer is used to within our end-to-end pipeline for multi-trajectory learning and detail the training regime.

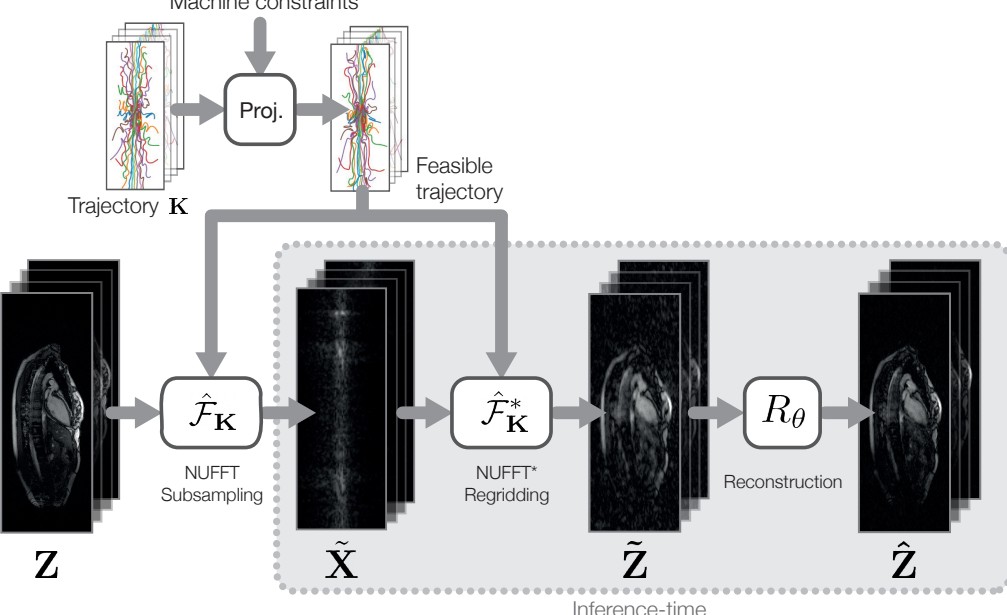

Figure 1: **Multi-PILOT Pipeline** Fully sampled frames $\mathbf{Z}$ are fed into our model, subsampled based on our trajectories and reconstructed.

### 2.1. Subsampling layer

Given a data sample composed of $n$ fully sampled frames, $\mathbf{Z} = [\mathbf{Z}_1, \mathbf{Z}_2...\mathbf{Z}_n]$, this layer returns a set $\tilde{\mathbf{X}} = \hat{\mathcal{F}}_{\mathbf{K}}(\mathbf{Z})$ of the $n$ subsampled frames in the frequency domain. We note the assumption our input is composed of a discrete set of frames is a simplifying assumption, more

applicable to the case of prospectively gated MRI acquisition. To perform subsampling, the layer maintains a representation of the acquisition trajectory $\mathbf{K} \in \mathbb{R}^{N_{\text{frames}} \times N_{\text{shots}} \times m}$ as learnable parameters, where $N_{\text{frames}}$ is the number of frames per data sample (along the temporal dimension), $N_{\text{shots}}$ is the number of RF excitations and $m$ is the number of sampling points within each shot. For each frame, we use the non-uniform FFT (NUFFT) algorithm (Dutt and Rokhlin, 1993) to obtain the subsampled image in the frequency domain at the non-Cartesian locations. To impose machine-related kinematic constraints for each sampling trajectory, we employ the projection algorithm proposed by (Chauffert et al., 2016; Radhakrishna and Ciuciu, 2023).

## 2.2. Regridding layer
We use the adjoint NUFFT (Dutt and Rokhlin, 1993) to transform our subsampled $k$-space data points into $n$ subsampled frames in the image domain, $\tilde{\mathbf{Z}} = \hat{\mathcal{F}}_{\mathbf{K}}^*(\tilde{\mathbf{X}})$.

## 2.3. Reconstruction model
The goal of the reconstruction model is, given the downsampled frames $\tilde{\mathbf{Z}}$ in the image domain, to output a set of reconstructed frames $\hat{\mathbf{Z}} = R_{\boldsymbol{\theta}}(\tilde{\mathbf{Z}})$, where $R_{\boldsymbol{\theta}}$ is the reconstruction network parametrized with learnable weights $\boldsymbol{\theta}$. Note that the network reconstructs a sequence of $n$ frames at once (collectively denoted as $\hat{\mathbf{Z}}$) given $n$ corresponding inputs (collectively denoted as $\tilde{\mathbf{Z}}$). To embody $R_{\boldsymbol{\theta}}$, we use the ACNN model proposed by (Du et al., 2021). ACNN is basically a U-Net (Ronneberger et al., 2015) model with attention and batch normalization layers applied throughout the pipeline, aimed at learning the optimal $k$-space interpolation for recovering data lost in undersampling the frequency domain. As mentioned above, an important aspect of dynamic MRI reconstruction is the opportunity to utilize data redundancy across different frames. ACNN addresses this need by learning the $k$-space interpolation for each frame based on several adjacent frames. Although ACNN was initially presented as a model for the undersampled reconstruction of static 3D MRI samples, in our work we adapt ACNN to reconstruct temporal sequences of two-dimensional images. It is important to emphasize that the principal focus of this work is not a specific reconstruction model; the proposed algorithm can be used with any differentiable model.

## 2.4. Training regime
The training of the trajectories and the reconstruction model is performed by solving the following optimization problem

$$\min_{\mathbf{K}, \boldsymbol{\theta}} \sum_i \mathcal{L}(R_{\boldsymbol{\theta}}(\hat{\mathcal{F}}_{\mathbf{K}}^*(\hat{\mathcal{F}}_{\mathbf{K}}(\mathbf{Z}_i))), \mathbf{Z}_i), \tag{1}$$

where the loss $\mathcal{L}$ (MSE in our experiments) is summed over a training set of fully sampled sequences $\mathbf{Z}_i$ each comprising $n$ frames. We emphasize that our goal was not to find an optimal loss function, having in mind that the proposed algorithm can be used with more complicated, possibly task-specific, loss functions.

The principal goal of training in such a multi-frame setting is to exploit the similarities across frames in order to achieve subsampling and reconstruction results superior to those of using a single subsampling trajectory, shared across all frames. As we later show in Section 3, naïvely feeding sequences of frames through our pipeline fails to achieve that. To our belief, this is due to the increased complexity of jointly optimizing a set of independent trajectories along with a reconstruction network. We found that the two training techniques described in the sequel were particularly beneficial to overcome this difficulty.

**Reconstruction resets** Jointly optimizing acquisition trajectories and the reconstruction model is a complicated optimization task, even within the setting of static MRI. Each optimization step over one component also induces an update in the other. This means, for example, that the current state of the reconstruction model is inherently dependent not only on the current subsampling trajectories but also, possibly, on much earlier states of the subsampling model. This dependency is not desired, as we mainly want the reconstruction model to perform the best only with the recent states of the acquisition model. Furthermore, this joint optimization is more susceptible to local minima. For this reason, in Multi-PILOT, we chose to reset the weights of our reconstruction model every $c$ training epochs, with $c$ being a hyper-parameter kept fixed throughout the training procedure. In Section 3, we show that this method is not only beneficial in the dynamic setting but also improves results in the static case. We believe that a similar technique can be applied in various joint sampling-reconstruction optimization efforts.

**Trajectory freezing** As previously stated, a key goal in multi-trajectory learning is to utilize cross-frame data redundancy and adjust learned trajectories to capture the unique features required for each frame. We observed that applying the optimization step over all trajectories at once complicates the task, as each trajectory is optimized under constant variations of the data acquired for neighboring frames. As a remedy, given some set of frames $\mathbf{Z} = [\mathbf{Z}_1, \mathbf{Z}_2...\mathbf{Z}_n]$, we propose to optimize each trajectory within our set of frames separately. Every trajectory is only optimized for some given number of epochs, and during that time all other trajectories are fixed. After optimizing each trajectory separately, we jointly fine-tune all trajectories for several epochs. The exact details of trajectory freezing method are further explicated in Appendix B. In our experiments, we restricted the freezing schedule to chronological order, deferring the investigation of the optimal optimization order for future work.

## 3. Experimental evaluation

### 3.1. Dataset

We used the OCMR dataset (Chen et al., 2020a), containing a total of 265 anonymized cardiovascular MRI (CMR) scans, both fully sampled and undersampled. Each sample consists of a set of a sequence of $384 \times 144$ 2D images with a variable number of frames. The data was acquired using Siemens MAGNETOM's Prisma, Avanto and Sola scanners. From this dataset, we only included a total of 62 scans containing fully-sampled multi-coil data.

Given the small amount of fully-sampled data, we augmented the data using vertical and horizontal flips, image re-scaling, and modulation of the frames with Gaussian masks to highlight varying image regions. Each of the augmentation operations was applied independently at random with a probability of 0.4 in each sample. Finally, we created our training, test, and evaluation samples by splitting all of the available videos into units of 8 frames. This was done to allow training on a larger set of data samples. We note that inference with a higher number of frames currently requires retraining the model. We aim to improve this in future work. After augmentation and splitting, 4170 samples were obtained. 80% of the data were allocated for training, 17.5% for testing, and 2.5% for the validation set.

### 3.2. Training setting

All experiments were run on a single Nvidia RTX A4000 GPU. Optimization in all experiments was done using the Adam optimizer (Kingma and Ba, 2014). For the reconstruction model, we applied dropout with a probability of 0.1 and used an initial learning rate of $10^{-4}$ with a decay of $5 \times 10^{-3}$ every 30 epochs. For trajectory learning, we used an initial learning rate of 0.2, decaying by a factor of 0.7 every 3 epochs. When applying reconstruction resets, we reset the reconstruction model every 35 epochs. When applying trajectory freezing, we optimized each trajectory for 35 epochs. When neither was applied, we executed each experiment for 315 epochs, so that all of our experiments ran for a total of 315 epochs (with or without resets and freezing). In all experiments batches of 12 samples each containing 8 $384 \times 144$ frames. Using this setting, our GPU memory consumption is up to 9.2GB, single epoch training time is around 13 minutes. Machine physical constraints for all experiments were Gmax = 40mT/m for the peak gradient, Smax = 200T/m/s for the maximum slew-rate, and dt = $10\mu$sec for the sampling time.

### 3.3. Quality metrics

For quantitative evaluation, we used peak signal-to-noise ratio (PSNR), visual information fidelity (VIF) (Sheikh and Bovik, 2006), and feature similarity indexing method (FSIM)(Zhang et al., 2011). PSNR measures pixel-wise similarity between images. VIF relies on statistical attributes expected to be shared between the target and the reconstructed image. FSIM compares images based on phase congruency and gradient magnitude that are known to be related to dominant features in the human visual system. According to the study of (Mason et al., 2019), VIF and FSIM were shown to be best correlated with the image quality scores assigned to a set of reconstructed MR images by expert radiologists. In spite of its popularity in other imaging and vision tasks, we chose not to include the structural similarity index measure (SSIM) within our metrics. This is because recent evaluations (Pambrun and Noumeir, 2015; Mason et al., 2019) as well as our own findings point to SSIM's inability to credibly represent similarity in medical imaging tasks.

### 3.4. Reconstruction results

In this section, we provide an ablation study (Table 1) showing that Multi-PILOT achieves superior per-frame image reconstruction compared to two baselines: 1. Golden Angle rotated stack of stars (GAR) $k$-space acquisition (Zhou et al., 2017) – a fixed (non-learned) $k$-space subsampling strategy, that according to the study of (Bliesener et al., 2020) provides state-of-the-art results in non-Cartesian dynamic MRI subsampling; and 2. PILOT (with a single trajectory applied to the acquisition of all frames) is brought as the trajectory learning baseline that most resembles ours, only without our proposed adaptations to the dynamic case. We additionally conduct an ablation study to learn the contribution of reconstruction resets and trajectory freezing.

In all of our experiments that include PILOT, we used the projection algorithm proposed by (Chauffert et al., 2016) to impose kinematic constraints, in spite of the fact that the original PILOT algorithm was penalty-based. This choice is dictated by the improved performance of projection-based version of PILOT, and also provides better grounds for comparison to Multi-PILOT which also utilizes the projection algorithm. In all experiments other than GAR, radial trajectory initialization was used. Golden Angle initialization was

|  | Learned traj. | Recon. Resets | Traj. Freeze | PSNR | VIF | FSIM |
|---|---|---|---|---|---|---|
| GAR | ✗ | ✗ | ✗ | $34.30 \pm 0.61$ | $0.772 \pm 0.011$ | $0.822 \pm 0.009$ |
| PILOT | Single | ✗ | ✗ | $35.87 \pm 0.74$ | $0.699 \pm 0.015$ | $0.8554 \pm 0.006$ |
|  | Single | ✓ | ✗ | $36.72 \pm 0.74$ | $0.705 \pm 0.013$ | $0.871 \pm 0.005$ |
|  | Multi | ✗ | ✗ | $34.06 \pm 0.62$ | $0.725 \pm 0.015$ | $0.806 \pm 0.011$ |
|  | Multi | ✗ | ✓ | $33.44 \pm 0.63$ | $0.684 \pm 0.01$ | $0.790 \pm 0.009$ |
|  | Multi | ✓ | ✗ | $37.18 \pm 0.72$ | $0.806 \pm 0.009$ | $0.875 \pm 0.008$ |
| Ours | Multi | ✓ | ✓ | $\mathbf{38.72 \pm 0.77}$ | $\mathbf{0.823 \pm 0.009}$ | $\mathbf{0.906 \pm 0.006}$ |

Table 1: **Reconstruction results comparison**.The proposed Multi-PILOT method (denoted as *Ours*) shows favorable reconstruction in all evaluation metrics.

also attempted, however empirically this initialization lead to sub-optimal results using our method.

Table 1 summarizes the performance of the compared reconstruction algorithms. The following conclusions are evident from the table. Firstly, the performance of the proposed method, trained with trajectory freezing and reconstruction resets surpasses both of our baselines according to all evaluated metrics. Our method achieves a 2 dB PSNR improvement, a 0.05 point VIF score improvement, and a 0.03 point FSIM score improvement compared to those achieved by our baselines. This suggests that independent per-frame trajectory learning achieves better reconstruction quality in dynamic MRI using our pipeline.

Secondly, the evaluation demonstrated the advantage of using reconstruction resets both for single and multi-trajectory learning. For single trajectory learning, we observed a 0.85 dB PSNR improvement. For multi-trajectory learning, the improvement exceeded 3.12 dB and 5.28 dB with and without trajectory freezing, respectively. Similar trends are manifested in the VIF and FSIM scores. The incorporation of trajectory freezing increased the metrics by 1.54 dB PSNR, 0.02 VIF points, and 0.03 FSIM points.

Thirdly, the evaluation shows that 'naïvely' learning multiple per-frame acquisition trajectories (without using reconstruction resets or trajectory freeze) achieves inferior reconstruction capabilities. This is true even in comparison to learning a single trajectory shared amongst all frames. We view this outcome as surprising since the solution to our optimization problem found by PILOT resides within the solution space of learning independent per-frame trajectories. We hypothesize that the reason for this result is the increased complexity of solving the optimization problem in its generalized multi-trajectory version. This assumption is supported by the noticeable improvement seen by incorporating reconstruction resets and trajectory freezing. Nonetheless, we believe that further investigation is required to elucidate this effect.

The favorable performance of our method is also shown in Figure 2. Compared to PILOT's reconstruction, Multi-PILOT exhibits significantly less imaging noise and artifacts. The corresponding acquisition trajectories and correlation with visual results are further explained in Appendix C.7. Additional visual results are presented in Appendix A.1.

### 3.5. Acquisition time minimization

In this section, we evaluate Multi-PILOT's potential in reducing MRI scan acquisition times by comparing the number of shots required for a certain reconstruction quality. As mentioned before, we sample every frame of the $k$-space using a constant pre-defined number of shots – each shot comprises a sequence of independently acquired 512 frequency samples.

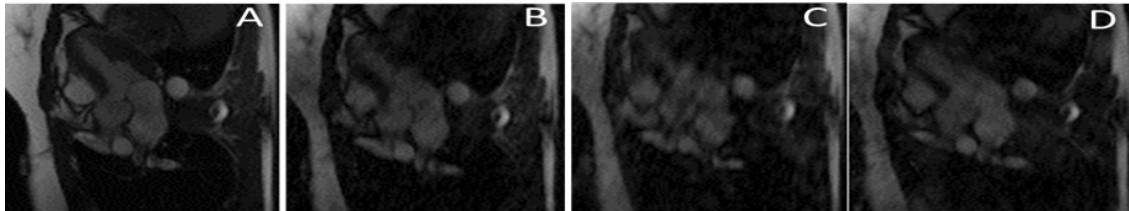

Figure 2: **Representative visual reconstruction results**. Fully sampled frame (A); reconstruction from undersampled data using Multi-PILOT (B), PILOT (C) and GAR (D).

| $N_{\text{shots}}$ | PILOT | | | Multi-PILOT | | |
|---|---|---|---|---|---|---|
| | PSNR | VIF | FSIM | PSNR | VIF | FSIM |
| 10 | $35.33 \pm 0.752$ | $0.629 \pm 0.014$ | $0.841 \pm 0.006$ | $36.32 \pm 0.70$ | $0.753 \pm 0.011$ | $0.855 \pm 0.008$ |
| 12 | $35.94 \pm 0.73$ | $0.672 \pm 0.014$ | $0.854 \pm 0.006$ | $37.35 \pm 0.72$ | $0.774 \pm 0.01$ | $0.878 \pm 0.008$ |
| 14 | $36.45 \pm 0.74$ | $0.685 \pm 0.012$ | $0.866 \pm 0.006$ | $37.16 \pm 0.72$ | $0.775 \pm 0.011$ | $0.873 \pm 0.009$ |
| 16 | $36.72 \pm 0.743$ | $0.705 \pm 0.013$ | $0.871 \pm 0.005$ | $38.72 \pm 0.77$ | $0.823 \pm 0.009$ | $0.906 \pm 0.006$ |

Table 2: **Acquisition time minimization.** Using $10-12$ shots, our method achieves reconstruction quality similar to that of our 16 shot baseline.

For our comparison, we explore the performance of our method and that of the next-best baseline: PILOT that uses a single trajectory shared across all frames, with reconstruction resets employed during training. We vary the number of shots used to sample the $k$-space.

Evaluation results are summarized in Table 2. Our primary conclusion is the potential of our method in reducing acquisition times. For example, in all metrics, a 12-shot version of Multi-PILOT achieves better reconstruction than a 16-shot PILOT, while 10-shot Multi-PILOT achieves comparable performance. This means that for a given required level of reconstruction, our method can use $25-35\%$ fewer shots/sample points compared to what our baseline would have to use. The results in Table 2 also support our results from Section 3.4 and show that our method provides substantially better reconstruction PSNR, VIF, and FSIM values compared to our baseline in additional settings. Visual reconstruction results are presented in Section A.2, and the depiction of corresponding learned trajectories can be found in Appendix C.

## 4. Conclusion

We investigated the task of dynamic MRI subsampling and restoration and discussed some of the challenges and unique considerations required when approaching this problem. As our solution to this problem, we proposed Multi-PILOT – an end-to-end pipeline for jointly learning optimal per-frame feasible $k$-space acquisition trajectories along with a multi-frame reconstruction model. Multi-PILOT is designed to address the distinct features of our problem within the dynamic setting (cross-frame data redundancy and complex optimization landscape). Our evaluation showed Multi-PILOT's potential for improving the reconstruction quality of dynamic MRI and reducing its acquisition time. We furthermore introduced reconstruction resets and trajectory freezing – two training methods that consistently and substantially improved reconstruction results within PILOT and Multi-PILOT training, and could be applicable to other subsampling and restoration pipelines.

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

## Appendix A. Visual Results

### A.1. Reconstruction Quality

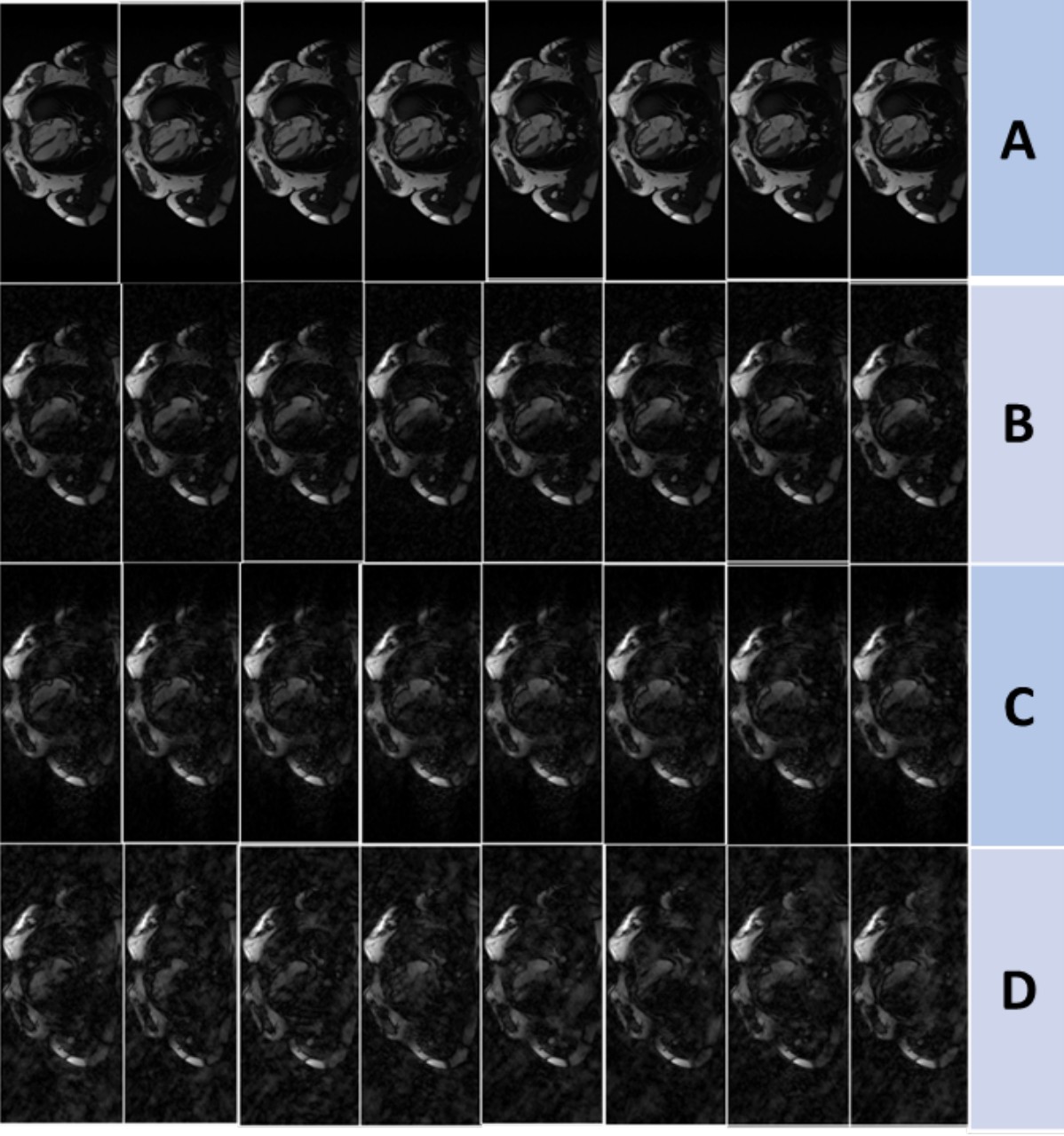

Figure A.3: **Visual reconstruction results** for Multi-PILOT (B), PILOT (C) and 'Naïve'
multi-trajectory learning (D), along with the ground truth frames (A). Our
method both reconstructs finer details of each frame and output substantially
cleaner images compared to PILOT and 'Naïve' multiple trajectory learning.

## A.2. Acquisition Time Minimization

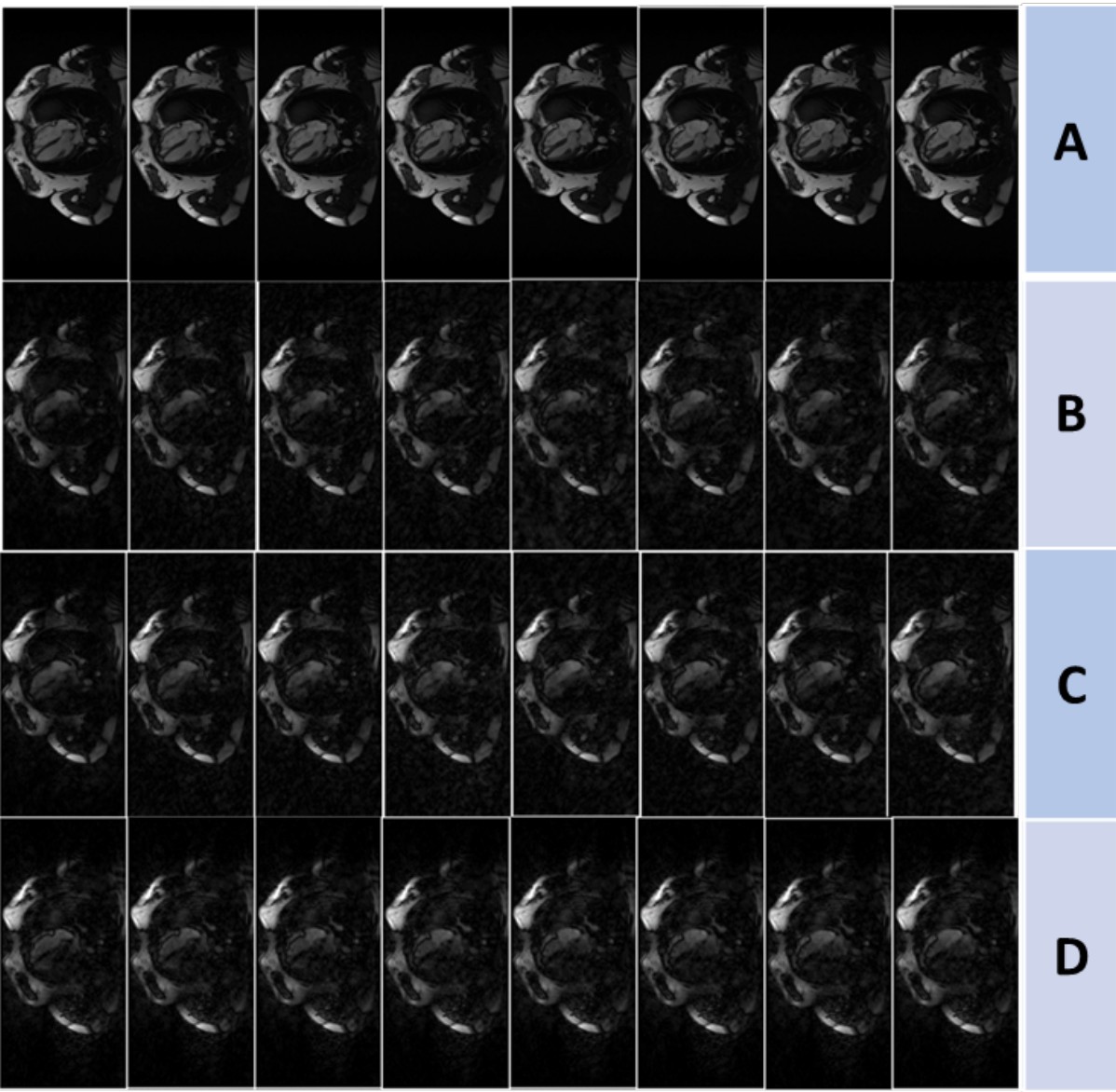

Figure A.4: **Visual reconstruction results for acquisition time minimization**. Ground truth frames (row A), Multi-PILOT with 10 (B) and 12 (C) shots produces cleaner reconstruction compared to PILOT with 16 shots (row D). Spatial resolution seems not to be compromised.

## Appendix B. Trajectory Freezing

In this section we elaborate on our trajectory freezing algorithm.

---

**Algorithm 1:** Trajectory Freezing Optimization

---

**Input:** Frame Sequence $Z_1, \ldots, Z_n$

1. Optimize only the reconstruction model and the first trajectory (matching frame $Z_1$). All other trajectories remain 'frozen'.;

2. **for** $i \leftarrow 2$ **to** $n$ **do**

    **for** $j \leftarrow 1$ **to** $i - 1$ **do**

        Set the previously learned trajectory $j$ as the acquisition trajectory for frame $Z_j$;

    **end**

    Initialize trajectory for $Z_i$ as the learned trajectory for $Z_{i-1}$;

    Optimize only the reconstruction model and the $i$th trajectory (matching frame $Z_i$). All other trajectories remain 'frozen'.;

**end**

3. Jointly optimize all learned trajectories and the reconstruction model (no frozen trajectories);

---

Stage 1 is done to initialize some starting point for our trajectory learning. In stage 2 we learn the trajectory for every frame based on all preceding frames. This is to allow the optimization to consider data already acquired by other frames. In stage 3 we jointly fine-tune our trajectories to allow each trajectory to be optimized based on information from all other acquisition trajectories.

As we show in 3, using trajectory freezing is optimal when combined with reconstruction resets. We reset the reconstruction network after stage 1 and after every running of stage 2.c.

## Appendix C. Trajectories

In this section, we present learned trajectories using PILOT and Multi-PILOT. In our training each data sample is consisted of 8 frames, therefore we start by initializing 8 identical radial acquisition trajectories.

Figure C.6 shows Multi-PILOT learns different acquisition trajectories across frames. This implies the optimal acquisition trajectories differ among frames along the temporal dimension, emphasizing the advantage of Multi-PILOT over applying static MRI pipelines in the dynamic setting.

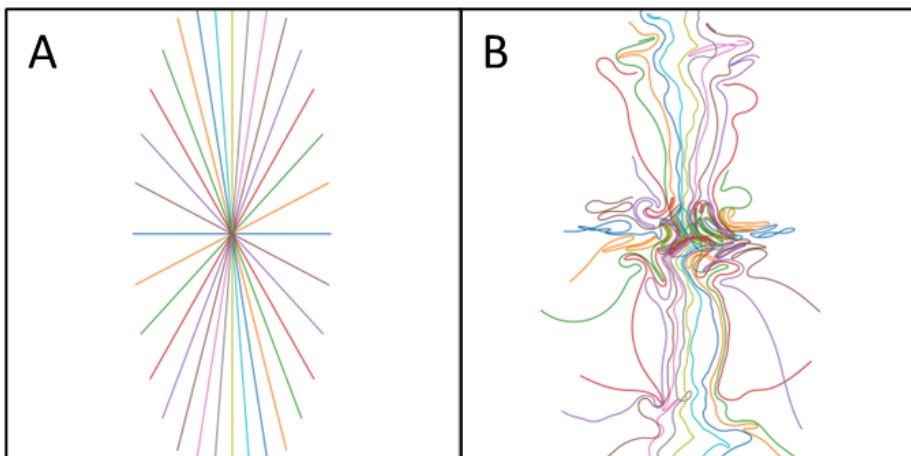

Figure C.5: **Trajectories shared across all frames**. A shows the radial initialization used to for every frame. B shows the learned shared trajectory obtained by applying PILOT in the dynamic setting

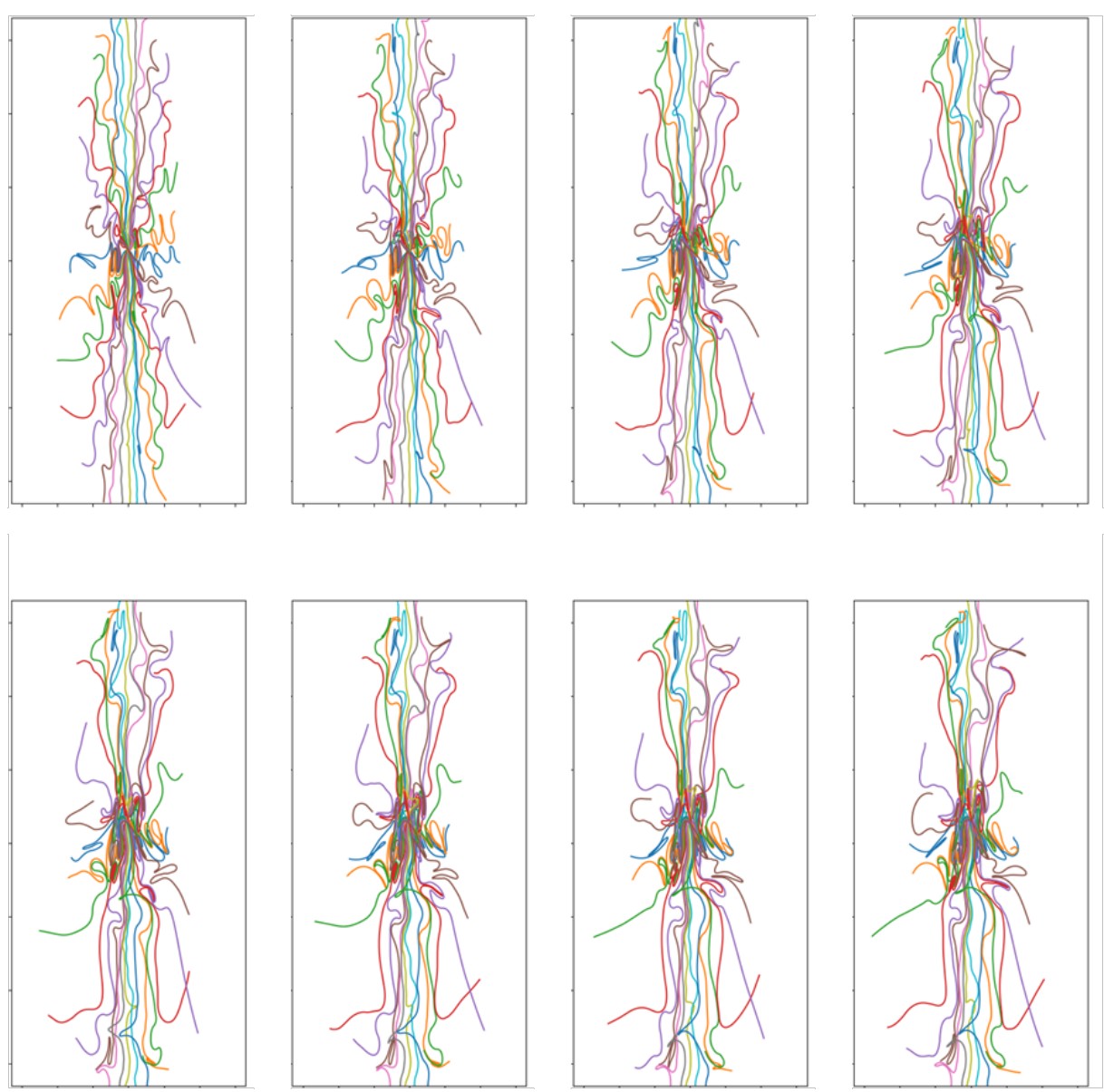

Figure C.6: **Learned per-frame trajectories using Multi-PILOT.**

We also present a more detailed explanation of 2.

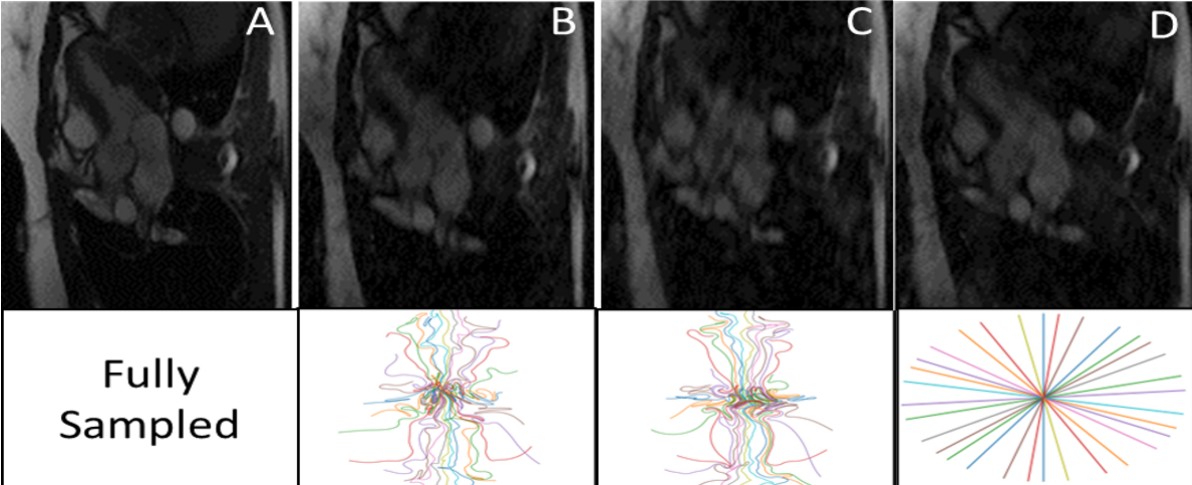

Figure C.7: **Representative visual reconstruction results**. Top row: fully sampled frame (A); reconstruction from undersampled data using Multi-PILOT (B), PILOT (C) and GAR initialization (D) (without trajectory optimization). Bottom row: corresponding trajectories with color-coded shots.

Figure C.7 shows the trajectories used to acquire the corresponding frames from 2. Many of the data in each frame is encapsulated at the center of the $k$-space. The learned trajectories show that while PILOT's single trajectory must allocate many acquisition points for capturing central frequencies, the Multi-PILOT trajectory can afford using trajectories more spread around the $k$-space, as data represented using the center frequencies are also captured by other frames. MultiPILOT does better in reconstructing the finer details compared to GAR initialization and PILOT reconstruction.

