# OpenReview forum: "Multi PILOT: Feasible Learned Multiple Acquisition Trajectories For Dynamic MRI"
_MIDL.io/2023/Conference — MIDL 2023 Poster_

### Official Review · Reviewer_sAYs · 2023-02-01

**Confidence:** 4
**Preliminary Rating:** 4
**Recommendation:** Poster

**Summary:**

The paper proposes a method called Multi PILOT to learn feasible sampling trajectories for dynamic MRI. The key idea is to learn per time-frame feasible trajectories, while still leveraging image correlation across time frames. The proposed method is evaluated using a public cardiac imaging dataset. The proposal is compared against the "naive" case where the sampling trajectory is not learned (i.e., radial sampling).

**Strengths:**

The paper is well-motivated and finding feasible sampling trajectories, especially for dynamic MRI, is a relatively unexplored field. The authors plan to make their code publicly available, which will help us in the scientific community to better understand the methods proposed in the paper. The proposed improvements seem to be responsible for the reconstruction improvements according to the authors' ablation studies.

**Weaknesses:**

Overall, I found the explanation of the method hard to follow. What makes a sampling trajectory feasible?

Regarding the data split in the experiment, there are 62 scans that result in 4170 samples. Was a subset of these scans used for testing? The authors say 2.5% of samples were used for testing, which would correspond to 1.55 scans. My concern is train and test contamination by having samples that come from the same scan both in the train and test sets.

My last concern is that the authors only compared their method against the "naive approach". Is the "naive" approach just using radial sampling or is it using PILOT in the static strategy?

**Deanonymize Review:**

no

**Detailed Comments:**

The sampling trajectories are hard to distinguish for the color blind.

**Paper Type:**

methodological development

**Questions To Address In The Rebuttal:**

I would ask the authors to clarify the data split they use to ensure there was no train/test contamination.

I would also ask the authors to clarify my other points in the "Weaknesses" in their revised manuscript.

---

### Official Review · Reviewer_urfr · 2023-02-02

**Confidence:** 5
**Preliminary Rating:** 1

**Summary:**

This paper describes a new method for optimizing (k,t) sampling trajectories for dynamic MRI.

There are several major problems: the paper lacks appropriate context, it's unclear how simulations were performed, and the validation study fails to compare against any of the sampling methods that are common for this scenario.

**Strengths:**

I am writing this statement to meet the length requirements of the review system.  I dislike that the review system requires me to waste time writing a lengthy response to this question.  See my detailed comments.

**Weaknesses:**

I am writing this statement to meet the length requirements of the review system.  I dislike that the review system requires me to waste time writing a lengthy response to this question.  See my detailed comments.

**Deanonymize Review:**

no

**Detailed Comments:**

It is hard to recommend publication of a paper that proposes a new (k,t) sampling scheme without making any effort to describe current approaches for (k,t) sampling, and which lacks comparisons against any of the prevailing methods.  The experimental results in this paper are inadequate because comparisons are only being made against sampling trajectories that are not well-suited to dynamic MRI.  It has been known for decades that using a fixed trajectory for all frames will lead to poor performance, so it is unacceptable to use that approach in the only "baseline" methods.

The concept of (k,t)-sampling for dynamic MRI was described by Xiang and Henkelman in the early 1990s.  Since that time, there has been a tremendous amount of work on designing good (k,t) trajectories.  This paper seems to be completely unaware of the long history of (k,t) trajectory optimization, and does not cite or compare against any of the standard approaches.  The following paper includes good examples of popular (k,t) trajectories, although these are by no-means complete due to the huge amount of work that has been done in this space:

Bliesener Y, Lingala SG, Haldar JP, Nayak KS. Impact of (k,t) sampling on DCE MRI tracer kinetic parameter estimation in digital reference objects. Magn Reson Med. 2020 May;83(5):1625-1639.

Suggesting that there is only limited work on (k,t) sampling design fails to acknowledge the major strides that have been made over the past 30 years, and it is shocking that this paper seems to have zero citations to papers about dynamic MRI acquisition or reconstruction.

The OCMR dataset contains complex-valued multi-channel k-space data.  It is unclear whether the authors are considering multi-channel sampling design, or if they've somehow simplified the situation to look at some form of single-channel sampling design.  This issue matters a lot, because optimal sampling patterns are quite different between single-channel and multi-channel sampling, as has been shown in previous work on trajectory optimization for static MRI

Haldar JP, Kim D. OEDIPUS: An experiment design framework for sparsity-constrained MRI. IEEE transactions on medical imaging. 2019 Feb 1;38(7):1545-58.

Gözcü B, Sanchez T, Cevher V. Rethinking sampling in parallel MRI: A data-driven approach. In2019 27th European Signal Processing Conference (EUSIPCO) 2019 Sep 2 (pp. 1-5). IEEE.

There are several different kinds of dynamic MRI acquisitions, including methods designed for real-time reconstruction, prospectively-gated acquisitions, and retrospectively-gated acquisitions.  Some of these approaches are incompatible with the proposed approach, and it's unclear which situation the proposed approach is designed for.  The paper has no discussion of this, and it's not clear whether the authors are aware of the different styles of acquiring dynamic MRI data.

The proposed approach is also based on an artificial model where the image is grouped into "frames" and each readout line occurs during one of the frames.  While this is a convenient model for reconstruction, it ignores the fact that the real dynamic image changes continuously over time, and every single readout line will be measured at a slightly different point in time.  (This may be mitigated if triggering is used in a prospectively-gated acquisition). This will lead to a lack of realism that may make the proposed approach appear to work better than it would with real data.  This is an important limitation that needs to be disclosed

I'm concerned about the way that experiments were performed.  Splitting into training, validation, and test sets should be done before data augmentation, not after. The test set should only contain original data, not any augmented data.  This will avoid problems where the augmentations might introduce unrealistic features.  The subjects used for the test set should also be distinct from the subjects used to create the training/validation sets.  The chances for overfitting rise dramatically of all three sets (training/validation/test) are derived by transforming a common core of original datasets than if they are created independently.  Based on the approach that was used in the paper, it is hard to trust the validity of the results.

The details of the experimental results are unclear.  What hardware constraints were used, and what assumptions were made about acquisition time?  The authors have used bSSFP data for simulation, but bSSFP generally requires a very short TR, which may be incompatible with the proposed trajectories.  This is especially a problem in section 3.5, this entire section may be disconnected from how real MRI acquisition works.

Figure 2 is badly scaled, we only care about what happens in the heart, but the scaling prevents us from seeing the heart clearly.

**Paper Type:**

methodological development

**Questions To Address In The Rebuttal:**

The paper suffers from serious flaws that I do not think are addressable with a rebuttal.  Fixing the problems would require a very substantial rewrite that also includes redoing some (possibly all) of the experiments.

---

### Official Review · Reviewer_uBeZ · 2023-02-04

**Confidence:** 4
**Preliminary Rating:** 5
**Recommendation:** Oral

**Summary:**

This paper describes an extension of PILOT, a method for joint learning of MRI acquisition trajectories and reconstruction networks, to the dynamic MRI setting. In particular, this work allows the learned trajectory to vary across frames in the series and allows the reconstruction network to incorporate multiple frames. This paper also introduces trajectory freezing (optimization of a single trajectory at a time) and reconstruction resets (resetting the reconstruction model weights every few training epochs) to improve the optimization of the entire system.

The paper compares Multi-PILOT to the standalone PILOT method with the learned acquisition and reconstruction scheme held constant across all frames and performs several ablation studies isolating the value of each component of the method.

**Strengths:**

- The paper is extremely clearly written and the details of the method are well-organized and easy to understand.
- The experiments are extremely well-designed; they isolate the improvements from every part of the system without overwhelming the reader.
- The method does appear to provide substantive improvements above the baseline Single-PILOT method. The experiment in Table 2 showing that reconstruction quality could be maintained with 25-35% fewer shots acquired is particularly convincing.

**Weaknesses:**

These weaknesses are minor in scope and I do not think any of these are reasons to reject the paper.
- The method is presented on a relatively small number of frames (8) — I believe that dynamic MRI sequences often are much longer than this.
- The presentation of the qualitative results/reconstructed images could be improved to make the improvement over other methods much easier to see.

**Deanonymize Review:**

no

**Detailed Comments:**

- Please comment on the scalability of the method to longer time series. I’m guessing memory use may be a constraint here due to the NUFFT in the network, and in general it would be good to add a comment to the main manuscript on the memory requirements/runtime of the proposed method.
- In the current formulation, would the method need to be retrained to be applied to dynamic time series of different lengths? The paper would benefit from some explicit discussion of this. A cool extension to this project may be to understand how to pick a set of acquisition/reconstruction “states” that in general would perform well across sequences of varying lengths, or to learn a strategy that dynamically decides when to learn a new acquisition/reconstruction state as the scan series progresses. (To be clear, I don’t think either of these are needed for acceptance of the current paper, they are just ideas for future work.)
- The visual reconstruction results could be made much clearer by including zoomed in patches of relevant regions where the reconstruction differs across methods and/or by including arrows that point out such regions.

**Paper Type:**

methodological development

**Questions To Address In The Rebuttal:**

- What are the memory requirements and training/inference clock time of the proposed method?
- In the current formulation, would the method need to be retrained to be applied to dynamic time series of different lengths?

---

### Meta-Review · Area_Chair_SLdF · 2023-02-26

**Recommendation:** Accept (Poster)
**Confidence:** 4

**Metareview:**

Having read the reviews and responses, I think that this publication has merit. I agree with `urfr` that augmentation of the test set is worrying at best, but I also agree with `uBeZ` that, after the rebuttal update, the comparison against GAR methods provides optimism.

I strongly recommend following the advice of `urfr` for experimental protocol. While I do not think that any of the augmentations are disqualifying/data poisoning, augmentation of a test set pollutes the measurement of error on "realistic" cases.

Overall however, I agree with `uBeZ`'s assessment of venue/placement.